# Identifying and Analyzing Low Energy Availability in Athletes: The Role of Biomarkers and Red Blood Cell Turnover

**DOI:** 10.3390/nu16142273

**Published:** 2024-07-15

**Authors:** Daisuke Suzuki, Yoshio Suzuki

**Affiliations:** 1Department of Biological Production Science, United Graduate School of Agricultural Science, Tokyo University of Agriculture and Technology, Fuchu 183-8509, Tokyo, Japan; dadadada0616@gmail.com; 2Graduate School of Health and Sports Science, Juntendo University, Inzai 276-1695, Chiba, Japan

**Keywords:** serum iron, hemoglobin, erythrocyte, erythropoiesis, female athlete triad, resting metabolic ratio

## Abstract

Low energy availability (LEA) is a growing concern that can lead to several problems for athletes. However, adaptation to LEA occurs to maintain balance over time, making diagnosis difficult. In this review, we categorize LEA into two phases: the initial phase leading to adaptation and the phase in which adaptation is achieved and maintained. We review the influence of LEA on sports performance and health and discuss biomarkers for diagnosing LEA in each phase. This review also proposes future research topics for diagnosing LEA, with an emphasis on the recently discovered association between red blood cell turnover and LEA.

## 1. Introduction

Low energy availability (LEA) in athletes is a growing concern. In the 1980s, an increased incidence of menstrual dysfunction was recognized in exercising women [1]. Subsequently, a link was suggested between menstrual dysfunction and energy balance [2]. These symptoms, characteristic of female athletes, were categorized as the Female Athlete Triad (Triad), in which the desire to achieve a prescribed weight goal leads to the development of disordered eating patterns, which in turn leads to amenorrhea and osteoporosis [3]. Subsequently, the validity of the concept that LEA causes the development of exercise-induced reproductive dysfunction was tested and confirmed in a monkey model [4]. In 2007, the American College of Sports Medicine (ACSM) redefined Triad to “refers to the interrelationships among energy availability, menstrual function, and bone mineral density” [5]. In 2014, the International Olympic Committee (IOC) Expert Working Group published a consensus statement to introduce a broader, more inclusive term, “Relative Energy Deficiency in Sport” (RED-S), which encompasses any impaired physiological function caused by relative energy deficiency in both female and male athletes [6]. Although Triad and RED-S are independent syndromes, they are commonly caused by LEA/energy deficiency [6,7].

As LEA has attracted widespread attention from the sport community, including researchers, athletes, sport leaders, and industry, there have been excellent review articles on this subject [8,9,10,11,12,13]. The International Olympic Committee (IOC) updated the consensus in 2018 [14] and 2023 [15]. Additionally, many review articles have been published within the last five years [16,17,18,19,20,21,22,23,24]. Areta et al. interpreted the history of the concepts of FAT and EA, summarized the biomarkers related to LEA reported until 2020 [18], and reviewed the impact of LEA on athletic performance [19]. Dipla et al. reviewed endocrine indicators associated with RED-S reported until 2019 [20]. Similarly, Melin et al. reviewed the effects of LEA on performance, as well as its physiological and psychological effects [21], and its impact on muscle strength, endurance, and injury and disease risk [22]. Popp et al. reviewed the effects of LEA on sex hormones and skeletal health [23]. Moreover, Shirley et al. reviewed the effects of LEA on reproduction, somatic cell maintenance, growth, and immunity from a life history theory perspective [24]. Various studies and reviews on these topics have been documented.

Despite many studies and reviewed results, LEA cannot be reliably diagnosed at this time due to the difficulty of accurately measuring energy intake and expenditure and the lack of clear signs that reflect LEA. This review proposes to understand LEA into two phases: a phase preceding adaptation and a phase wherein adaptation is achieved and maintained. When confronted with an energy deficit caused by LEA, the human body attempts to restore equilibrium by regulating metabolism, with adaptation to a specific EA occurring after a prolonged period. In this aspect, we discuss the biomarkers for diagnosing LEA at each stage and suggest areas for future research. In addition, a brief review of the impact of LEA on athletic performance and health is provided.

## 2. Metabolic Adaptation to Low Energy Availability

Energy availability (EA) is defined as “the dietary energy left over and available for optimum function of body systems after accounting for the energy expended from exercise” [15]. Energy availability is calculated by subtracting exercise energy expenditure (EEE) from dietary energy intake (EI), expressed as kg per kg fat-free mass (FFM) per day (kcal/kg FFM/d) [15]. When EEE is large and total energy expenditure exceeds EI, the remaining EA becomes insufficient to maintain physiological functions, which is LEA [15]. The threshold for LEA in females is discussed to be 30 kcal/kg FFM/d, while in males, it is debated to be lower (e.g., ~9 to 25 kcal/kg FFM/day) and has not reached a consensus [15]. One of the main reasons for the difficulty in understanding the threshold is the lack of a pronounced sign of LEA in males, like amenorrhea in females. The lack of pronounced signs for LEA is basically because the human body has plasticity and metabolic adaptation occurs.

When the intake of a nutrient decreases rapidly, the body tries to maintain balance. This response has been well studied in the context of nitrogen balance studies. In response to a dietary change to a protein-free diet, urinary nitrogen excretion decreases rapidly between 3 and 8 days and adjusts to a statistically similar level of daily excretion between 10 and 14 days [25]. Nitrogen losses, such as urinary and fecal excretion and skin shedding while adapted to a nitrogen-free diet, reflect obligatory metabolic demands. Changing from a high-protein diet to a low-protein diet does not change the obligatory metabolic demand for nitrogen, but it does decrease the adaptive metabolic demand from 109 mgN/kg/d to 38 mgN/kg/d by day 14 [26]. This adaptation to lower levels of protein intake would be expected to be accompanied by a decrease in resting metabolic rate (RMR) since protein turnover requires 15–20% of RMR when both energy and protein are adequate [27,28].

Similarly, when energy intake decreases, RMR decreases as “adaptive thermogenesis” or “metabolic adaptation” occurs to maintain balance for the survival of the individual [29]. Notably, decreased RMR has been reported after approximately 12 weeks of weight loss on a low-calorie diet [30]. Koehler et al. reported that exercising women with exercise-associated amenorrhea had lower RMR than women with ovulatory cycles [31]. However, it is not as clear as with nitrogen how many days this adaptation will take. The problem is compounded by the fact that the human body stores glycogen and triglycerides as energy sources, whereas nitrogen stores little. When non-obese healthy adults with a body mass index (BMI) of 26.1 ± 2.7 kg/m^2^ underwent a 43-day hunger strike, weight, BMI, and triceps skinfold continued to decrease from baseline, and the decreases were significant from day 18 to day 31, but not significant at days 31 and 43 [32]. In this case, it may be possible to evaluate that the adaptation took 31 days. However, there have been no reports on the time course to reach adaptation when LEA occurs with increased EEE under constant EI conditions. In addition, the number of days to adapt to LEA is likely to be influenced by energy storage at baseline; there have been no reports on this aspect.

Therefore, it remains unclear how many days are required to achieve adaptation to LEA. Nevertheless, even nitrogen with little or no storage in the human body takes about 10 days to adapt to low nitrogen availability; thus, LEA adaptation is expected to take 2 weeks or more.

## 3. Influence of Low Energy Availability on Athletic Performance

Questionnaire-based studies have reported that LEA is associated with disrupted training schedules in elite athletes [33], poor training response, endurance performance, and impaired judgment, coordination, and concentration in female athletes [34], as well as lower power-to-weight ratios in male cyclists [35]. Woods et al. found that trained cyclists who underwent a 4-week block of intensive training with reduced energy intake had impaired regenerative capacity and poorer performance on a 5000-m track [36].

Melin et al. reviewed the literature, categorizing LEA exposure into short-term, medium-term, and long-term, and concluded that severe LEA exposure can impair athletic performance due to direct/indirect health effects, hormonal changes, and suboptimal levels of energy substrates (i.e., muscle glycogen) [22]. In contrast, Areta summarizes that LEA does not directly impair athletic performance [19]. For example, while LEA reduces muscle protein synthesis [37,38], resistance exercise has been shown to rescue skeletal protein synthesis to the same extent as when energy is adequate [37,39,40]. A meta-analysis found that energy deficiency prevented lean mass gains but did not suppress the increase in muscle strength after an 8–20-week resistance training program [41]. Elite and world-class athletes with acute and chronic energy deficits achieved aerobic capacity and performance commensurate with their requirements [42,43,44,45,46,47]. Based on these reviews, Areta concludes that there is no definitive evidence to support whether energy deficiency is beneficial or detrimental to performance and that more research is needed [19].

Tokuyama et al. reported unintended weight loss resulting from LEA at a rugby camp [48]. Murphy et al.’s meta-analysis showed that an energy deficit of 500 kcal/d suppressed the increase in lean mass [41]. Therefore, LEA should be avoided if one of the goals is weight gain or maintenance.

## 4. Influence of Low Energy Availability on Health

The most striking effect of LEA is weight loss. Areta et al. [18] have shown from studies with reduced energy availability that EA deficiency is proportional to weight loss (%) in the first 3–5 days, and this short-term change is due to a decrease in skeletal muscle glycogen [49,50,51] and the water bound to it [52,53]. Observations on hunger strikers indicate that the decrease in carbohydrates (mainly glycogen with its associated water) occurs initially, followed by a decrease in protein and fat [54]. During the 43-day hunger strike, weight reduced by an average of 8.9% in the first 11 days, while the 20-day reduction from day 11 to day 31 averaged 9.0%, during which time the triceps skinfold decreased from an average of 17.8 mm to 11.9 mm [32]. The lower rate of weight loss after day 11 compared to before is likely due to the use of energy-dense fats as fuel.

Since death occurs when approximately 19% of body protein and 70–94% of body fat stores are lost [54], weight loss itself has significant health consequences. However, a meta-analysis has shown that combat athletes typically undergo a process called “making weight,” characterized by rapid weight loss of about 10% in the days before a competition, but rapid weight gain of about 11% in the 3 to 32 h after weigh-in [55]. Another meta-analysis showed significantly increased tension, anger, and fatigue and decreased vitality in judo athletes who lost weight rapidly, but no specific disease onset or irreversible disability was reported [56]. Nonetheless, to promote safer practices, it is recommended that the time between weigh-in and competition be shortened and that unhealthy weight-loss techniques be prohibited [55].

The ACSM’s position on Triad noted the multiple health consequences of Triad, which include psychological problems associated with eating disorders and medical complications in the cardiovascular, endocrine, reproductive, skeletal, gastrointestinal, renal, and central nervous systems, reproductive problems associated with amenorrhea, and loss of bone mineral density [5]. The IOC Consensus Statement 2023 on RED-S lists a range of potential health outcomes resulting from problematic LEA, including impaired reproductive function, bone health, gastrointestinal function, energy metabolism regulation, hematologic status, urinary consistency, glucose and lipid metabolism, mental health, neurocognitive function, sleep, cardiovascular function, skeletal muscle function, growth and development, and immunity [15]. These symptoms of Triad and RED-S are quite undesirable, but not completely identical, and there is no conclusive symptom to diagnose LEA.

While weight loss due to LEA, if not severe, may be recovered in a relatively short period with no obvious sequelae, some of these symptoms may take a long time to recover from or may not be expected to recover adequately. For example, bone density typically peaks between the ages of 20 and 30 and then declines [57,58,59]. It is not clear whether bone density can be restored to the same level as with adequate energy intake if it is reduced by LEA during growth. Therefore, prolonged LEA should be avoided due to its serious health consequences, which may include irreversible sequelae.

## 5. Importance of Diagnosing Low Energy Availability

LEA can cause unintended weight loss even in the short term (as discussed in Section 2) and is believed to contribute to various health issues in the long term [15]. Particularly, it may have irreversible effects on bone density (as discussed in Section 4). However, changes in bone density occur gradually, and it may take several years for imaging techniques to detect these changes [60]. Therefore, to prevent the drawbacks of unintentional LEA, it should be assessed before health outcomes occur.

The excellent review by Areta et al. summarizes the endocrine, metabolic, and physiological effects reported through 2020 [18]. Similarly, Dipla et al. offer a thorough overview of endocrine changes associated with RED-S through 2019 [20]. While these reviews are not systematic, they represent a comprehensive synthesis of important studies conducted through 2020 on biomarker changes caused by LEA. 

When EA is reduced, the human body attempts to modulate metabolism to maintain balance [29]. As discussed in Section 2, this adaptation may take 2 weeks or more, and there may be differences in biomarker levels between the early phase when the body is responding to LEA and regulating its metabolism toward adaptation, and the phase when the body has fully adapted to LEA and its metabolism is maintained. Therefore, in the following sections, we will organize the findings to date based on whether adaptation to LEA has not been achieved (defined as acute LEA) or has been achieved and maintained (defined as chronic LEA) (Figure 1).

However, LEA cannot be diagnosed from the symptoms, because the symptoms of Triad and RED-S are different and inconclusive. In addition, because the EA threshold has not reached a consensus, LEA cannot be determined from an energy perspective, even if EI, EEE, and energy balance are accurately measured. Therefore, in the following discussion, the EA when the energy balance is maintained at the optimal level is considered to be the “sufficient level”, and when it is less than that, it is considered to be LEA. Thus, there may be a case of LEA where the apparent energy balance is maintained.

## 6. Diagnosing Acute Low Energy Availability

As discussed in Section 2, it is likely that more than two weeks will be required to adapt to LEA. Therefore, in this and the next section, we discuss biomarkers that can be used to diagnose acute and chronic LEA within two weeks and beyond, respectively. Biomarkers that have been reported to reflect low energy availability are summarized in Table 1. However, the biomarkers listed in Table 1 do not point in the same direction in all studies. The discrepancy may be due to the lack of consensus on the threshold for LEA, as well as the subjects’ baseline energy stores and other factors.

In their reviews, Areta et al. [18] and Dipla et al. [20] identify the following biomarkers altered by short-term LEA. Endocrine responses include decreases in triiodothyronine (T3) [61,62,63,64,65], IGF-1 [63,64], insulin [61,62,63,64,65,66,67], leptin [62,65,68,69,70], and Luteinizing Hormone (LH) [71], and increases in cortisol [71] and aldosterone [63]. Blood metabolites show a decrease in glucose [31,61,62,66,67] and free fatty acids [63] and an increase in beta-hydroxybutyrate [37,64,65,68]. LEA decreases skeletal muscle protein synthesis in both men and women [37] and skeletal muscle glycogen levels in men [50,51]. Furthermore, a female-specific response includes a disruption of the LH pulse [61,62,64,65,66] and an increase in GH [62,66,72] and FSH [70]. In women, a decrease in bone formation markers and an increase in bone resorption markers have been suggested [68,69,73], though results in men have been inconsistent [68]. While a decrease in testosterone has been proposed as a male-specific response, it is rarely reported in short-term LEA [74].

**Table 1 nutrients-16-02273-t001:** Biomarkers that have been reported to reflect low energy availability.

Gender	Acute LEA	Ref.	Chronic LEA	Ref.
Both	Triiodothyronine (T3) ↓	[61,62,63,64,65,75,76]	Triiodothyronine (T3) ↓	[77,78,79,80]
	IGF-1 ↓	[63,64,75]	IGF-1 ↓	[77,78,81]
	Leptin ↓	[62,65,67,68,69]	Testosterone ↓	[78,80,81]
	Luteinizing hormone (LH) ↓	[62,71]		
	Insulin ↓	[61,62,63,64,65,66,67,82]		
	Glucose ↓	[61,62,65,66,67]		
	Protein synthesis ↓	[37]		
	Growth hormone (GH) ↑	[62,65,66,72,82]		
Female	Follicle-stimulating hormone (FSH) ↑	[66]	Leptin ↓	[80,83,84,85]
	Osteogenic markers ↓	[68,69,73]	Ghrelin ↑	[84,86]
	Bone resorption markers ↑	[68,69,73]	Estradiol ↓	[77,85]
	Beta-hydroxybutyrate ↑	[61,62,65,76]	Progesterone ↓	[77]
	Forearm resting blood flow	[76]	Luteinizing hormone (LH) ↓	[83]
Male	Muscle glycogen ↓	[50,51]	Cholesterol ↓	[78]
	Hemoglobin ↓	[75,87,88]	Insulin ↓	[78]
	Cortisol ↑	[63,82]	Thyroxine (T4) ↓	[82]
	IGFBP-1 ↑	[63]		
	Aldosterone ↑	[63]		
	Free fatty acids ↓	[63]		
	Iron ↓ & transferrin saturation ↓	[48,77,78,79,80]		
	Testosterone ↓	[82]		
	Sleep efficiency, N3 stage proportion ↓	[89]		
	Wake after sleep onset ↑	[89]		

The ↑ and ↓ represents increase and decrease, respectively.

A series of RCTs conducted after 2020 by Jurov et al. suggested that a 14-day reduction in EA results in changes in iron status in addition to the previously mentioned hormonal alterations. In trained elite endurance athletes, a 25% reduction in baseline EA by increasing energy expenditure while maintaining energy intake was observed to reduce hemoglobin, serum iron, and IGF-1 levels [87]. Similarly, when EA was reduced by 50%, a significant decrease in hemoglobin, IGF-1, and T3 levels was observed [75]. Additionally, a stepwise reduction of EA at three levels (25%, 50%, and 75%) over 10–14 days showed that several biochemical parameters responded differently to different levels of EA reduction [88]. Consistent with their previous study, a 25% to 50% reduction in EA resulted in decreases in hemoglobin, serum iron, and IGF-1. However, these markers of iron status were not altered by a 75% reduction in EA. T3 and testosterone did not change with a 25–50% reduction in EA, but did decrease with a 75% reduction [88]. In addition to the above studies, Tokuyama et al. reported a decrease in serum iron and transferrin saturation after one week of LEA during a summer training camp for male college rugby players [48]. Therefore, iron status markers may be indicators of early signs of reduced EA.

Recent studies have added new potential biomarkers. While the association between sleep state and LEA has been mentioned [15,33,90], Saidi et al. found that sleep parameters (sleep efficiency, promotion of N3 stage, and wake after sleep onset) indicate the difference between optimal energy availability and low energy availability [89]. Hutson et al. also found that resting forearm blood flow decreased in response to LEA, along with known changes in T3 and β-hydroxybutyrate [76].

Red blood cells (RBCs) make up about 50% of blood and 4% of body weight. The lifespan of RBCs is about 120 days, with a daily turnover of approximately 0.04% of body weight (equivalent to about 30 g for a 75 kg person). However, since erythropoiesis requires substantial energy, RBC turnover is likely suppressed to maintain energy balance in response to LEA. Erythropoiesis is regulated by erythropoietin, the primary erythropoietic factor. Reduced erythropoietin levels in the blood have been reported to promote the selective lysis of young RBCs [91]. Conversely, old RBCs are phagocytosed by macrophages in the spleen and hepatic sinusoids. This phagocytosis is antagonistically regulated by phosphatidylserine and CD47 [92]. 

Recently, the strict time course of calcium and Na/K pump activity decay and PIEZO1-mediated calcium permeation have been proposed to be involved in RBC lifespan, although volume and density changes during capillary transit are transient and recover within a few days [93]. The menstrual cycle is also reported to influence RBC levels in women [94]. Since LEA can affect any of the above processes of RBC turnover, even for a short period, research into the relationship between RBC status and LEA is warranted.

To summarize the above findings regarding the biomarkers associated with LEA, from the perspective of diagnostic use, cut-off values should be established to evaluate differences from adequate EA. Methods to diagnose short-term responses, such as the OGTT for diagnosing glucose tolerance, should also be developed. Additionally, the reliability of markers associated with RBC turnover, which have recently been suggested to be relevant, should be further investigated.

## 7. Diagnosing Chronic Low Energy Availability

As described in Section 4, we defined the phase in which adaptation to LEA has not been achieved as acute LEA and the phase in which it has been achieved and maintained as chronic LEA. However, determining the state of adaptation is challenging. EA is defined as energy intake minus exercise energy expenditure, expressed per kilogram of FFM [15], and includes RMR. Torstveit et al. evaluated the measured ratio to predicted RMR in male endurance athletes and reported its association with cortisol concentration and the testosterone:cortisol ratio [95]. RMR can be a good indicator because it is expected to decrease with adaptation to LEA. 

Research on chronic LEA biomarkers is still limited compared to short-term LEA. The reviews by Areta et al. [18] and Dipla et al. [20] list the following biomarkers that reflect LEA for more than 14 days. Levels of IGF-1 [77,78] and T3 [77,78,79] decrease in both men and women. Unlike short-term LEA, several studies have consistently reported a decrease in testosterone in men [78,81,96,97]. In women, long-term LEA induces decreases in estradiol and progesterone [77] and increases in ghrelin [86].

In addition to the literature cited in the two reviews [18,20], decreased leptin has been consistently observed in women with hypothalamic amenorrhea due to strenuous exercise [83], in normal-weight women competing in fitness sports after a 4-month fat loss diet [80], in synchronized swimmers after 4 weeks of intensified training [84], and in runners during a training overload phase [85]. These studies also reported decreased LH [83], T3 [80], testosterone [80], and estradiol [80,85], and increased ghrelin [84]. Decreased testosterone has also been reported in male runners with the exercise-hypogonadal male condition [81]. In contrast, in healthy young male volunteers, decreased insulin and testosterone and increased cortisol and GH were observed after 5 days of a 20-day field exercise program, but these hormones returned to baseline levels by the end of the program, whereas decreased thyroxine was observed [82].

To diagnose chronic LEA, cut-off values should be established, and a method should be developed to assess short-term responses, similar to the OGTT used to diagnose glucose tolerance.

Chronic LEA is also expected to affect biomarkers involved in red blood cell turnover. A study of active men by Hennigar et al. showed that a 55% reduction in EA over 28 days reduced hemoglobin levels [98] This decrease in hemoglobin (anemia), which does not respond to iron administration, may be due to the suppression of erythropoiesis by LEA and should be evaluated in this context. Additionally, in situations where erythropoiesis is chronically suppressed and hemoglobin levels are balanced by prolonging erythrocyte lifespan, a decrease in the percentage of neoplastic erythrocytes and an increase in the percentage of old erythrocytes are expected. Comparing these percentages to optimal levels can help detect chronic LEA. Further study of the association between chronic LEA and RBC turnover is warranted.

## 8. Conclusions

It is not clear whether LEA affects athletic performance in athletes. However, LEA may cause unintentional weight loss in the short term and various health problems in the long term, including potentially irreversible consequences for bone health. Therefore, unintentional LEA should be avoided in both the short and long terms. 

However, the body attempts to maintain balance through “adaptive thermogenesis” or “metabolic adaptation” when EA is reduced, making LEA difficult to diagnose. Consequently, methods are needed to diagnose LEA in the dynamic state before and after adaptation is achieved. Diagnostic methods should include establishing cut-off values for biomarkers associated with LEA. Additionally, further studies are expected on the association between LEA and erythrocyte turnover, which has recently been identified as relevant.

## Figures and Tables

**Figure 1 nutrients-16-02273-f001:**
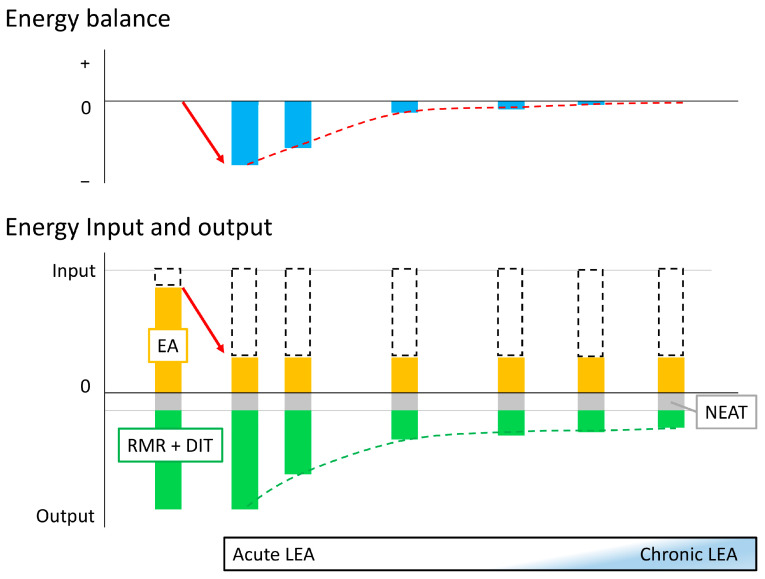
Concept of acute and chronic low energy availability Initially, energy availability (EA; orange box) is sufficient to maintain energy balance (leftmost bar). When energy intake and non-exercise activity thermogenesis (NEAT; gray box) are constant and exercise energy expenditure (white dashed box) increases (red arrow), low energy availability (LEA) occurs, and energy balance becomes negative: this is the onset of acute LEA (second left bar). As this state persists, adaptation occurs and resting metabolic rate (RMR) and diet-induced thermogenesis (DIT) (cumulative green box) gradually decrease (green dashed line), and the body attempts to maintain energy balance (red dashed line). Eventually, adaptation to the LEA state is achieved to maintain apparent energy balance while suppressing RMR and DIT (rightmost bar). However, it is not yet clear how many days it takes to achieve adaptation and what conditions are sufficient to diagnose adaptation as achieved (chronic LEA).

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
