# Peer review of "Identifying and Analyzing Low Energy Availability in Athletes: The Role of Biomarkers and Red Blood Cell Turnover"

_nutrients, 2024, doi:10.3390/nu16142273_

Round 1

Reviewer 1 Report

Comments and Suggestions for Authors

The terminology used could more closely reflect the IOC statement on RED-S i.e. adaptable and problematic LEA.

Also not all studies show the same direction of change in the biomarkers proposed, it would be good to have more discussion on these biomarkers and why the findings are not consistent between studies.

Ensure correct use of REDs and LEA terms and be consistent with the IOC statement on the abbreviations.  

It was recognised by some that menstrual dysfunction 

Line 33: Change The International Olympic Committee (IOC) consensus was established in 33

2023  to the latest consensus statement

Comments on the Quality of English Language

Some improvements required throughout

Author Response

Thank you for taking the time to review this manuscript. 
Your comments have really helped to improve the manuscript.
Please find the detailed responses attached WORD file.
The corresponding revisions/corrections are highlighted with the track change in the resubmitted manuscript.

Reviewer 2 Report

Comments and Suggestions for Authors

Line/Section

Comments

Introduction

Lines 21-30

1.       Please define Low Energy Availability, and describe the causal role of LEA and reproductive health.

  • Williams NI, Helmreich DL, Parfitt DB, et al. Evidence for a causal role of low energy availability in the induction of menstrual cycle disturbances during strenuous exercise training. J Clin Endocrinol Metab 2001;86:5184–93.

2.       For this review, it is critical authors note that there are 3 consensus statements that need to be integrated into the introduction, you currently have 2 of the 3 (the male athlete triad as 2 parts); and make sure all 3 are properly defined per the consensus statements. It is also important to note the Triad and REDS are independent syndromes and one did not replace the other, therefore, discuss this independently with the commonality being LEA/energy deficiency. For the Female Athlete Triad do not abbreviate as FAT, this has a negative connotation, consider simply using “Triad”.

  • De Souza MJ, Williams NI, Nattiv A, Joy E, Madhusmita M, Loucks A, et al. Misunderstanding the Female Athlete Triad: Refuting the IOC Consensus Statement on the Relative Energy Deficiency in Sports (RED-S). Brit J Sport Med. 2014;48(20):1461-1465.

·         Nattiv, Aurelia, et al. "The Male Athlete Triad—A Consensus Statement from the Female and Male Athlete Triad Coalition Part 1: Definition and Scientific Basis." Clinical Journal of Sport Medicine: Official Journal of the Canadian Academy of Sport Medicine (2021).

·         Fredericson, Michael, et al. "The Male Athlete Triad—A Consensus Statement from the Female and Male Athlete Triad Coalition Part II: Diagnosis, Treatment, and Return-To-Play." Clinical Journal of Sport Medicine 31.4 (2021): 349-366.

·         Mountjoy M, Ackerman KE, Bailey DM, et al. 2023 International Olympic Committee’s (IOC) consensus statement on Relative Energy Deficiency in Sport (REDs). British Journal of Sports Medicine 2023;57:1073-1098.

3.       Authors state the 2023 statement now recognizes LEA now in males, this is incorrect, this was first mentioned in the 2014/2015 REDS consensus statement and again in the 2021 Male Athlete Triad Consensus Statement. Please provide accurate resources and definitions throughout.

  • Mountjoy M, Sundgot-Borgen J, Burke L, Carter S, Constantini N, Lebrun C, Meyer N, Sherman R, Steffen K, Budgett R, Ljungqvist A. The IOC consensus statement: beyond the Female Athlete Traid – Relative Energy Deficiency in Sport (RED-S). Brit J Sport Med. 2014;48:497. doi: 10.1136/bjsports-2014-093502.
  • Mountjoy M, Sundgot-Borgen J, Burke L, Carter S, Constantini N, Lebrun C, Meyer N, Sherman R, Steffen K, Budgett R, Ljungqvist A. Authors’ 2015 additions to the IOC consensus statement: Relative Energy Deficiency in Sport (RED-S). Brit J Sport Med. 2015,49(7):417-420.

4.       Integrate more current literaturre on LEA (for a review paper, we need to know the authors have genuinetly reviewed all the literature, these references are only within the last 3 years, there are many more.

·         Torres-McGehee TM, Emerson DM, Flanscha-Jacobson A., Uriegas NA, Moore, EM, Smith AB. Examination of Energy Availability, Mental Health, and Sleep Patterns among Athletic Trainers. Journal of Athletic Training. 2023;58(9):788-795. https://doi.org/10.4085/1062-6050-0547.22

·         Bacon, Caitlin Nicole. "The Prevalence of At-Risk Female Athletes for Developing Low Energy Availability in Division I and III Collegiate Athletics." (2024).

·         Sharples, Alice, et al. "Risk of low energy availability, eating disorders and food insecurity amongst development female rugby league players." The Journal of Sports Medicine and Physical Fitness (2024).

·         Smith AB, Gay J, Arent SM, Sarzynski M, Emerson D, Torres-McGehee TM.   Examination of Prevalence of the Female Athlete Triad Components among Competitive Cheerleaders: International Journal of Environmental Research and Public Health. 2022; 19(3):1375. https://doi.org/10.3390/ijerph19031375

·         Salamunes, Ana Carla C., Nancy I. Williams, and Mary Jane De Souza. "Are menstrual disturbances associated with an energy availability threshold? A critical review of the evidence." Applied Physiology, Nutrition, and Metabolism 49.5 (2024): 584-598.

·         Kommi K, Saifi MA, Khanna PG. 530 BO46 – Risk for low energy availability, disordered eating and sleep disturbance in female football players British Journal of Sports Medicine 2024;58:A68-A69.

·         Melin, Anna K., et al. "Direct and indirect impact of low energy availability on sports performance." Scandinavian journal of medicine & science in sports 34.1 (2024): e14327.

·         Lye, Jamie Ching Ting, et al. "Low Energy Availability and Eating Disorders Risk: A Comparison between Elite Female Adolescent Athletes and Ballet Dancers." Youth 4.2 (2024): 442-453.

·         Sim, Alexiaa, et al. "Original Investigation: Manipulating energy availability in male endurance runners: a randomised controlled trial." Applied Physiology, Nutrition, and Metabolism ja (2024).

·         Buch, T., et al. "Risk of low energy availability and level of nutrition knowledge in recreational trail runners in Aotearoa/New Zealand." Proceedings of the Nutrition Society 83.OCE1 (2024): E42.

Lines 33

The statement on the IOC being established in 2023, is incorrect, the first REDS statement was published in 2014, updated in 2015 and 2023.

Lines 34-44

Literature also includes examination of LEA with or without an eating disorder risk. Physiological changes may also be present due to the eating disorder pathology (changes in biomarkers). If you are being inclusive in all consensus statements regarding LEA, then this needs to be included. Numerous references are provided above.

Lines 45

Provide more information on the reliability of assessing LEA, what are the barriers and limitations to the methods used.

Overall Introduction

The background information on energy availability, low energy availability, Female and Male Athlete Triad and REDS, needs to be further defined by the authors. For example, what are the “cutoff” for LEA and adequate LEA, and then in your metabolic section below, describe the literature that supports these metabolic changes with LEA. These are independent syndromes and need to be presented as independent syndromes or this will be confusing to the reader. It is also important to clearly define the medical consequences associated with each, so the other sections flow better.

Metabolic Adaptation to LEA

As mentioned above, authors should consider adding literature on how LEA effects these metabolic changes. The actual section does not mention LEA until the last sentence.

Influence of LEA on Athletic Performance

The review in the section is lacking literature, there is a lot more literature (below are just a few articles). Would like to see a more comprehensive review for this section.

·         Logue, Danielle, et al. "Low energy availability in athletes: a review of prevalence, dietary patterns, physiological health, and sports performance." Sports medicine 48 (2018): 73-96.

·         Melin, Anna K., et al. "Direct and indirect impact of low energy availability on sports performance." Scandinavian journal of medicine & science in sports 34.1 (2024): e14327.

·         Stables, Reuben G., et al. "Acute fuelling and recovery practices of academy soccer players: implications for growth, maturation, and physical performance." Science and Medicine in Football 8.1 (2024): 37-51.

·         Logue, Danielle M., et al. "Low energy availability in athletes 2020: an updated narrative review of prevalence, risk, within-day energy balance, knowledge, and impact on sports performance." Nutrients 12.3 (2020): 835.

·         Loucks, Anne B. "Low energy availability in the marathon and other endurance sports." Sports Medicine 37 (2007): 348-352.

·         Melin, Anna K., et al. "Energy availability in athletics: health, performance, and physique." International journal of sport nutrition and exercise metabolism 29.2 (2019): 152-164.

Influence of LEA on Health

In the introduction, the authors introduce both the Triad and REDs, it is highly recommended in this section to provide a review of the literature for both syndromes and/or at least provide information on the commonalities and differences.

Importance of diagnosing Acute and Chronic LEA

Authors need to be clear on the definitions of LEA, these are different in the REDs and Triad consensus statements. Then use supporting literature for diagnosing and possibly challenges for measuring and diagnosing LEA

Overall

The overall manuscript seems to be targeted toward REDs, however because the Triad and REDS are two separate syndromes, it’s important for the authors to clearly identify which one they are reviewing and if they are reviewing only one (REDS) then this needs to be clear and not confused with the Triad. The introduction needs the most work for clarity of the work that will be outlined. The sections also need more information on the actual Biomarkers and Red Blood Cell turnover. When discussing performance and diagnosis, this information seems to get lost. In the diagnosis section, both REDS and Triad have assessments that can also be used in the clinic, consider adding those in as well.

Comments on the Quality of English Language

No comments on the quality of the English language

Author Response

(The authors gave the same response as above.)

Reviewer 3 Report

Comments and Suggestions for Authors

Low energy availability (LEA) in athletes, initially linked to menstrual dysfunction in women, is now recognized as a broader issue affecting both genders, termed Relative Energy Deficiency in Sport (RED-S). Despite extensive research, diagnosing LEA remains challenging.

LEA leads to metabolic adaptations, notably a reduction in resting metabolic rate (RMR) as the body attempts to maintain balance. This adaptation can occur within about 12 weeks and the adjustment processes relating to the utilization of energy sources appear to occur in succession. The impact of LEA on athletic performance is debated: some studies show impaired performance and recovery, while others find no direct performance impairment if energy deficiency continues.

Health consequences of LEA include significant weight loss, primarily from muscle glycogen depletion, and long-term effects like impaired bone density, which may not fully recover. The IOC 2023 consensus lists risks such as reproductive, bone, and cardiovascular issues.

Diagnosing LEA involves identifying biomarkers affected by energy deficiency. Short-term LEA indicators include changes in hormones like triiodothyronine (T3), IGF-1, insulin, and leptin, and blood metabolites like glucose and free fatty acids. Chronic LEA biomarkers include reductions in IGF-1, T3, and testosterone in men, and estradiol and progesterone in women. Iron status and red blood cell turnover are also affected.

The work of Suzuki D and Y describes the metabolic adaptation processes in detail. It is interesting to note the temporal sequence of the adaptation processes; they distinguish acute low energy availability from a chronic form.  I have the following comments on the submitted manuscript:

1.       Introduction
Please add some remarks on the prevalence of LEA. As the popularity of women’s female physique athletes is inceasing every year the incidence of LEA is becoming more frequent. It would also be useful to know which sports are particularly risky in this respect as LEA is frequently overlooked.

2.       Paragraphs 2 and 4 are not well separated. When it comes to metabolic adaptation processes, bone metabolism should also be mentioned here.  However, this is described in lines 131-134.
On the other hand, paragraph 4 (lines 104-109) rather mentions metabolic changes.
Think about renaming this section 2 to „Metabolic and physiologic adaptations…“
LWA may also modulate regional blood flow and vascular function (DOI: 10.1007/s00421-024-05497-0). Also, the sleep quality (doi: 10.3390/nu16050609) and
fatigue level seem to be influenced (DOI: 10.1139/apnm-2024-0037). Further adaptions as well as symptoms (e.g. gastrointestinal or immunological) have been described. Influence of LEA on heath then could be more focused on health relevant sequelae

6. The authors describe possible changes in erythropoiesis. However, it remains unclear which diagnostic measures can be derived from this. If the authors could provide an assessment or recommendation here, this would be helpful for readers (what lab work should be requested in addition to iron status markers, if any?).

Author Response

(The authors gave the same response as above.)
